# High-contrast, synchronous volumetric imaging with selective volume illumination microscopy

Thai V. Truong [1,2,5 ✉], Daniel B. Holland[1,5], Sara Madaan[1,3,5], Andrey Andreev[1,3], Kevin Keomanee-Dizon [1], Josh V. Troll[1], Daniel E.S. Koo[1,3], Margaret J. McFall-Ngai[4] & Scott E. Fraser [1,2,3 ✉]

Light-field fluorescence microscopy uniquely provides fast, synchronous volumetric imaging by capturing an extended volume in one snapshot, but often suffers from low contrast due to the background signal generated by its wide-field illumination strategy. We implemented light-field-based selective volume illumination microscopy (SVIM), where illumination is confined to only the volume of interest, removing the background generated from the extraneous sample volume, and dramatically enhancing the image contrast. We demonstrate the capabilities of SVIM by capturing cellular-resolution 3D movies of flowing bacteria in seawater as they colonize their squid symbiotic partner, as well as of the beating heart and brain-wide neural activity in larval zebrafish. These applications demonstrate the breadth of imaging applications that we envision SVIM will enable, in capturing tissue-scale 3D dynamic biological systems at single-cell resolution, fast volumetric rates, and high contrast to reveal the underlying biology.

[1] Translational Imaging Center, University of Southern California, Los Angeles, CA 90089, USA. [2] Molecular and Computational Biology Section, University of Southern California, Los Angeles, CA 90089, USA. [3] Department of Biomedical Engineering, University of Southern California, Los Angeles, CA 90089, USA. [4] Pacific Biosciences Research Center, University of Hawaii at Manoa, Honolulu, HI 96822, USA. [5] These authors contributed equally: Thai V. Truong, Daniel B. Holland, Sara Madaan. ✉email: tvtruong@usc.edu; sfraser@provost.usc.edu

Understanding dynamic biological processes requires volumetric imaging tools that can faithfully image across hundreds of microns in three dimensions (3D) with cellular resolution within time scales as short as milliseconds. Conventional imaging approaches based on sequentially collecting signal from one point (confocal microscopy), one line (line-confocal), or one plane (light-sheet) at a time[1,2] are often not fast enough to faithfully capture the relevant dynamics without distortion, as different parts of the 3D sample are observed at different times. Light-field microscopy (LFM; Fig. 1a) meets this challenge by capturing an extended sample volume in a single snapshot, enabling synchronous volumetric imaging[3–5]. LFM records the extended light field coming from the sample space on a two-dimensional (2D) camera by positioning a micro-lens array at the image plane, and moving the camera to the focal plane of the micro-lens array. This permits the camera to capture information from the volume that extends above and below the native focal plane. Computational reconstruction is used to solve the inverse problem, reconstructing the image of the 3D sample volume from the recorded 2D image, sacrificing resolution for dramatically enhanced z-depth coverage[3–5]. LFM conventionally employs wide-field illumination, exciting sample regions beyond the volume of interest (Fig. 1b), thus generating background signal that reduces the contrast of both the recorded 2D image and the final reconstruction. The limited contrast of conventional LFM has substantially limited its utility for imaging dynamic 3D biological tissues.

Taking inspiration from selective plane illumination microscopy (SPIM; also known as light-sheet microscopy)[6], which achieves low-background and high-contrast imaging by illuminating only the optical plane of interest (Fig. 1b), we reasoned that we could enhance the contrast of LFM by illuminating only the volume of interest. We thus created selective volume illumination microscopy (SVIM) by preferentially illuminating the volume of interest and then capturing the resulting fluorescence with light-field detection. SVIM reduces background, increases contrast, and produces an overall higher-quality reconstruction of the sample, while preserving the synchronous volumetric imaging capability of LFM.

## Results

**Overview of SVIM instrument.** Our SVIM instrument combined selective volume illumination and LFM modules with an existing custom-built SPIM[7], permitting direct comparison between SVIM, conventional wide-field LFM, and SPIM imaging of the same specimen (Methods section, Supplementary Fig. 1, Supplementary Table 1). SPIM provided slower, but higher-resolution, "ground truth" images against which to judge the other imaging modalities. To achieve selective volume illumination, in either 1-photon (1p) or 2-photon (2p) excitation, we implemented galvanometer-based rapid scanning of the specified volume multiple times within a single camera exposure time[8,9], providing micron-level control over the spatial extent of the selected volume (Methods section, Supplementary Note 1). Our design of the light-field detection module drew upon previous efforts[3–5], and the light-field image reconstruction followed the 3D deconvolution approach[4] using publicly available software[5]. Supplementary Table 2 provides imaging and reconstruction parameters for all presented results. As previously described theoretically and experimentally[3,4], LFM image reconstructions are affected by non-uniform resolution and grid-like artifacts centered around the native focal plane, both of which were present in our results. SVIM performed as expected from the optical parameters used[4], achieving a nominal maximum resolution of ~3 μm laterally and ~6 μm axially, as approximated by the full-width half-maximum (FWHM) of sub-diffractive fluorescent beads, over a volume of $440 \times 440 \times 100$ (x, y, z) μm³ (Supplementary Fig. 2). Our SVIM implementation provides a simple path for conventional SPIM instruments to be upgraded to SVIM.

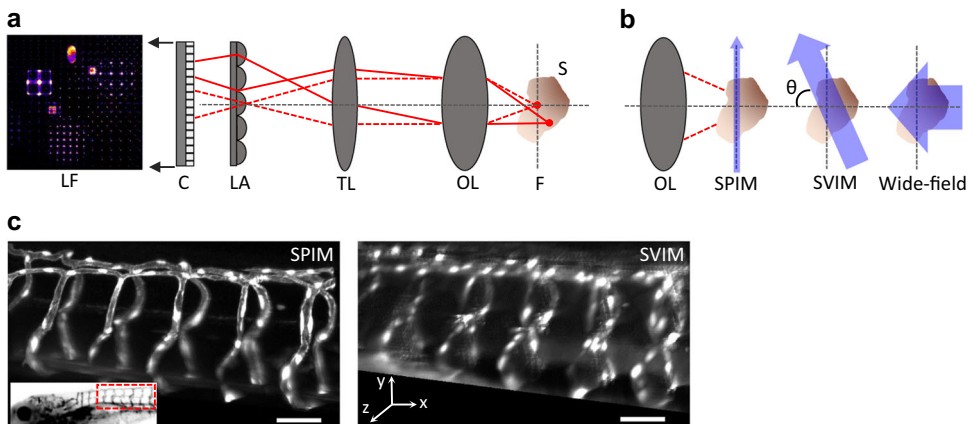

**Fig. 1 Selective volume illumination microscopy enhances LFM for the synchronous imaging of 3D samples. a** LFM is a simple extension of a conventional microscope, which produces a magnified image of the sample (S) from the native focal plane (F) to the image plane (IP) using an objective lens (OL,) and tube lens (TL). LFM places a micro-lens array (LA) at the IP, encoding 3D image information into a 2D light-field image (LF), which is captured by a planar detection camera. This permits LFM to synchronously capture information at z-positions above and below F; the 3D image of the sample is reconstructed from the LF image, based on knowledge of the optical transformation. **b** SVIM improves LFM by selectively illuminating the volume of interest within the sample. This decreases background and increases contrast when compared to wide-field illumination of the entire sample. SVIM was implemented through the use of light-sheet (SPIM) illumination that is scanned axially, so that the thin sheet of excitation is extended into a slab. In our work, the SVIM illumination axis was orthogonal to the detection axis ($\theta = 90^0$), but the benefits of reduced background can be obtained by using illumination from a different angle, and/or by employing non-linear optical effects to selectively excite the volume of interest. **c** SPIM and SVIM 3D images of the trunk vasculature of 5 dpf zebrafish larva reveal the compromises between resolution and volumetric imaging time. SPIM offers higher resolution but requires the collection of 100 sequential images to cover the 100-μm-depth z-stack; SVIM captures the same 3D volume in a single snapshot, two-orders-of-magnitude faster, but with lower resolution. Transgenic animal, *Tg(kdrl:GFP)*, had its vasculature fluorescently labeled with green fluorescent protein (GFP). Inset shows the approximate location of the imaged volume along the trunk of the zebrafish larva. Scale bars, 50 μm.

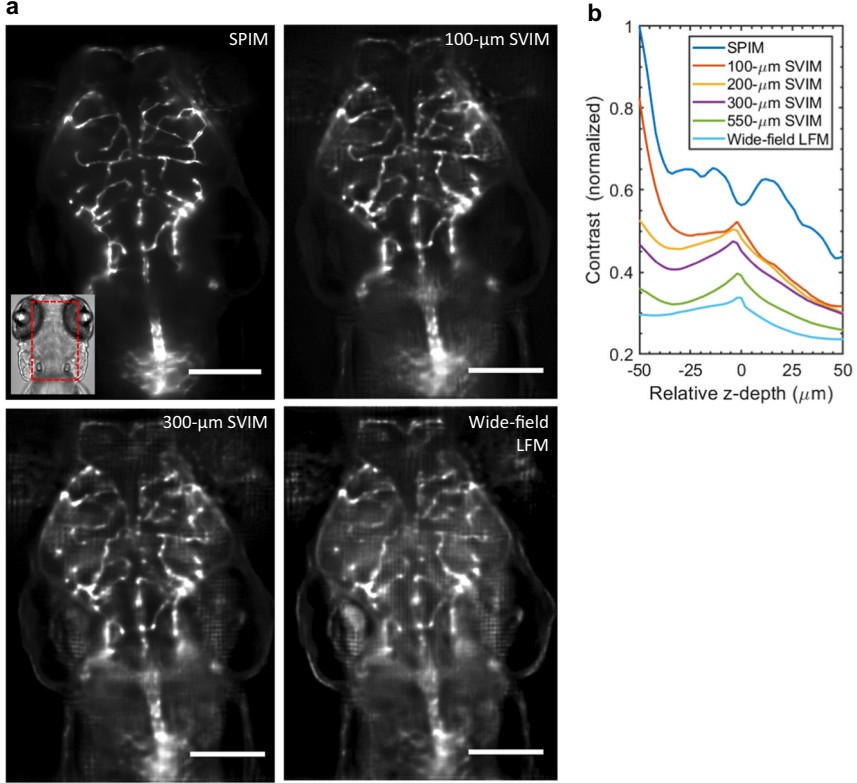

**Fig. 2 Higher contrast achieved by SVIM. a** SVIM images of the cranial vasculature improve in contrast as the depth (axial extent) of the illumination volume is decreased. Images are averaged-intensity $z$-projections of the same 40-µm thick sub-volume, centered at ~170 µm into the head of a 5 dpf zebrafish larva. The SVIM image quality progressively approaches the performance of SPIM as the axial extent of the illumination is reduced to 300 µm or 100 µm, far exceeding the image contrast obtained with wide-field LFM. Inset shows the approximate location of the imaged volume, in context of the zebrafish head. **b** Quantitative comparison of image contrast, defined as the normalized standard deviation of the pixel values (Methods section), comparing LFM, SPIM, and SVIM of different SVI extents from **a**. SVIM of smaller extents yielded increasingly better contrast, approaching the performance of SPIM. The contrast of SPIM showed the intrinsic contrast variation of the 3D sample, coupled with the expected contrast decay for increasing imaging depth. The local increase in contrast seen for the SVIM and LFM cases around $z = 0$ µm came from grid-like artifacts from the light-field reconstruction centered around the native focal plane, a known feature of LFM in general[3,4]. Scale bars, 100 µm.

**SVIM imaging performance**. The capabilities and compromises of SVIM are demonstrated by its single-snapshot capture of the entire depth of the trunk vasculature of a live larval zebrafish (Fig. 1c). Compared to the $z$-stack assembled from 100 higher-resolution SPIM snapshots, SVIM captured faithfully the 3D structure of the green fluorescent protein (GFP)-labeled vasculature. SVIM demonstrated modest reductions in resolution, but its single-snapshot acquisition offered two-orders-of-magnitude greater $z$-depth coverage and enhanced imaging speed, even after normalization for the number of resolvable voxels captured (Supplementary Table 2, Supplementary Note 2).

SVIM enhances image quality compared with wide-field LFM, as seen in the 3D images of the cranial vasculature of the same live zebrafish larva, with the illuminated volumes varying from 100-µm axial extent to wide-field illumination of the entire animal (Fig. 2a, Supplementary Fig. 3). Wide-field LFM produced the lowest-quality image with the highest background. SVIM produced progressively better-quality and higher-contrast images as the $z$-extent was reduced, approaching the ground truth images achieved by SPIM when the SVI was 100 µm (Supplementary Fig. 3g). Measurements of the quantitative image contrast (Methods section) show SVIM's progressively increased performance as the illumination extent was confined to smaller volumes (Fig. 2b). The decreased contrast from background is consistent with our simulations (Supplementary Fig. 4), where increased levels of Poisson noise applied to the raw light-field images

resulted in decreased contrast in the reconstructions. The higher contrast of SVIM mitigates against resolution-degrading effects of background noise, resulting in a better effective resolution even though SVIM and wide-field LFM utilize the same detection optics. This was demonstrated in imaging ~5-µm-diameter blood vessels where SVIM achieved up to 35% improved FWHM over wide-field LFM (Supplementary Fig. 5).

**SVIM enhances imaging of biological components moving in 3D**. The synchronous volumetric imaging capability and enhanced contrast of SVIM is ideal for imaging dynamic systems, where components undergo fast motion in 3D space. We employed SVIM to image the bacterial flows in seawater, surrounding the light organ of a Hawaiian bobtail squid, *Euprymna scolopes*, while it was selectively colonized by the bacteria *Vibrio fischeri*[10]. The squid–bacteria symbiosis is an important model for understanding the effects of fluid flow during interactions between bacteria and epithelial surfaces[11]. Previous 2D measurements of the bacterial flow field[12] inadequately captured the 3D flows around the light organ. SVIM offered dramatically better image quality as compared to wide-field LFM (Fig. 3a, b). SVIM removed most of the background that severely compromised wide-field LFM, which came from the excitation of nearby auto-fluorescent tissues. In SVIM, the fluorescence of individual bacteria could be clearly imaged and tracked (Fig. 2b, Supplementary Movies 1, 2 and 3), yielding the position (Fig. 3c) and speed (Supplementary

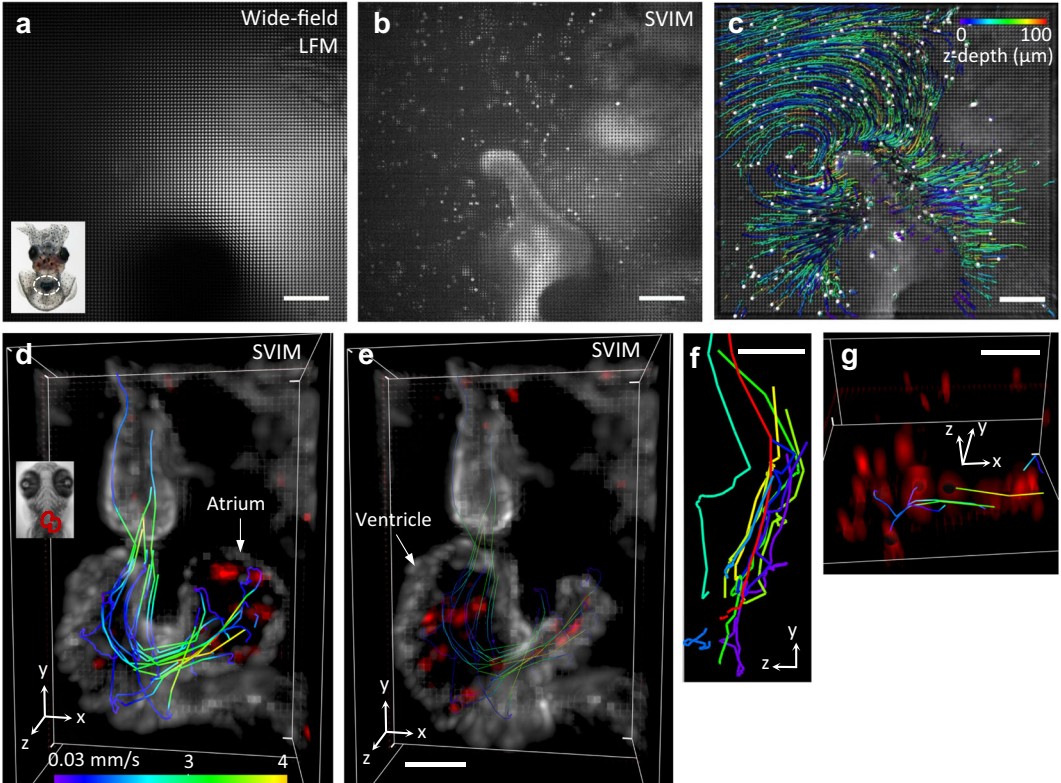

**Fig. 3 SVIM enables fast, high-contrast, volumetric imaging of live biological systems. a–c** Imaging the bacterial flow around the light organ of a juvenile squid. Raw light-field images recorded with conventional wide-field illumination yielded excessive background **a**, whereas SVIM, with a selectively illuminated volume of 100 μm, reduced this background and enhanced the contrast to allow localization of individual bacteria **b**. Inset shows squid with the light organ region highlighted by the dashed oval. **c** Quantitative flow trajectories tracked from the reconstructed SVIM data, color-coded for z-depth. Non-uniform 3D flow patterns were observed throughout the imaged volume. Images were collected at 20 volumes s$^{-1}$, with a volume ~600 × 600 × 100 μm$^3$ (depth). **d–g** Imaging the motions of the beating heart wall and moving blood cells. A volume of ~250 × 150 × 150 μm$^3$ (depth) in a live 5 dpf zebrafish larva was captured at 90 volumes s$^{-1}$. Transgenes labeled the endocardium (rendered white) and blood cells (rendered red), *Tg(kdrl:eGFP, gata1:dsRed)*. Inset in **d** highlights the position of the heart within the animal. The captured beating heart is shown in 3D-rendered views at two representative time points during the cardiac cycle: **d** the atrium was at its fullest expanded extent, followed by **e** when the blood had been pumped into the enlarged ventricle. Representative blood cell flow trajectories were manually tracked and quantified (color of the trajectories in **d**, **e**, **g** depicts blood cell speed; Methods section). **f** Maximum projection image along the x-axis of several representative flow trajectories highlights the substantial component of blood flow along the z-direction. To aid visualization, clipping planes in the yz plane were used to cut out the atrium and parts of the ventricle. Color-coding of the blood cell tracks in **f** is only for visual identification. **g** Perspective view of the blood cells demonstrates the achieved single-cell resolution, notably along the z-direction. Circular voids within several blood cells mark the cells whose trajectories were tracked and quantified. Scale bars, (**a–c**) 100 μm, (**d–g**) 50 μm.

Fig. 6c) of each bacterium, and a full quantitative description of the flow field in 3D.

We further tested the performance of SVIM by imaging the motions of the live beating heart of an intact zebrafish larva (Fig. 3d–g, Supplementary Movies 4 and 5), important for understanding how dynamic cellular and fluid motions contribute to the heart development[13,14]. SVIM captured the beating motion of the heart walls and the trajectories of blood cells, with single blood cell resolution, at 90 volumes s$^{-1}$ over the entire heart (Fig. 3d, e, Supplementary Movies 4 and 5). This represents a volume coverage rate five times larger than an optimized plane illumination approach could achieve[15]. Compared to wide-field LFM, SVIM achieved images with 50% and 10% better contrast of the heart wall and blood cells, respectively (Supplementary Fig. 7). SVIM synchronously captures dynamic cardiac behaviors, free of potential artifacts that could arise in other beating heart imaging methods that rely on specific presumptions about the nature of the heart motions[7,16–18]. Thus SVIM could be ideally suited for 3D imaging of arrhythmias and other transient, non-periodic heart beating behavior in studies involving genetic, physical, or pharmacological perturbations.

**SVIM enhances brain-wide functional neuroimaging.** Many recent developments of LFM have focused on functional neuroimaging[5,19–24], despite the challenges presented by the relatively slow image reconstruction algorithms, and the reduced image quality of LFM compared with state-of-the-art neuroimaging techniques[2]. This is because the extraordinary imaging rate possible with LFM could be game changing for the simultaneous recording of large number of spatially distributed neurons. We tested if the enhanced contrast of SVIM would improve the recording of neural activity in larval zebrafish, as assayed through a genetically encoded calcium indicator expressed in all of its neurons (Fig. 4, Supplementary Figs. 8 and 9). The enhanced contrast from the reduced background of SVIM, in both 1p and 2p excitation modes (Supplementary Fig. 8), enabled better performance than wide-field LFM in recording the calcium transients that reflect the firing of single neurons across the zebrafish brain, capturing up to fourfold more neurons during spontaneous brain activity (Methods section; Fig. 4d–f). While both 1p and 2p excitation SVIM offered improved contrast, they present different compromises. Excitation with 1p offers simplicity and fast volumetric imaging rates, as demonstrated with LFM in general[5,19–24].

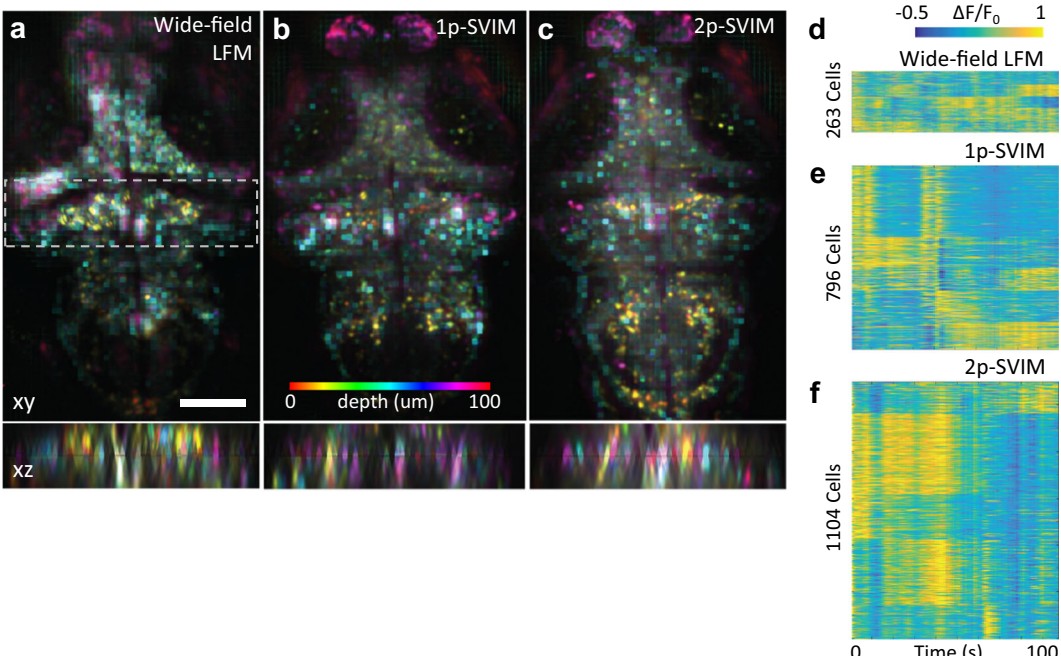

**Fig. 4 Functional neuroimaging with SVIM.** Functional imaging of a 5 dpf larval zebrafish with pan-neuronal fluorescent calcium indicators, *Tg(elavl3:H2b-GCaMP6s)*. Spontaneous brain activity, over a volume ~600 × 600 × 100 µm³ (depth), was recorded at 1 volume s⁻¹, with SVIM, in either 1- or 2-photon excitation mode (1p-SVIM or 2p-SVIM, respectively), or conventional 1p wide-field LFM. Cellular-resolution representations of active neurons were found with standard methodology based on spot segmentation of the time-domain standard deviation of the 3D time-series data (Methods section). **a–c** Images shown are depth color-coded *xy*- or *xz*-projections, of the time-domain standard deviation projection of the recorded brain activity over a time window of 100 s. Colored ellipsoids represent active neurons. Dashed box in the *xy*-projection image represent the region that produces the corresponding *xz*-projection image. Activity traces of segmented neurons are shown in **d–f**, revealing that the most number of neurons were found with 2p-SVIM (1104 cells), then with 1p-SVIM (796 cells), both of which were several-fold higher than with conventional wide-field LFM (263 cells). Scale bars, 100 µm.

The lower 2p excitation cross section limits the imaging rate of 2p-SVIM; however, we found that 2p excitation led to better contrast and a larger number of resolved active neurons (Fig. 4e, f, Supplementary Fig. 8). This detection of more active neurons, enabled by the improved contrast, results not only from the reduced tissue autofluorescence of 2p excitation, but also from additional mechanisms specific to our application. First, 2p laser light preserves its spatial profile better than 1p light as it penetrates deeper into the sample[9], due to the reduced scattering at longer wavelengths. Thus, the scanned selectively illuminated volume is more precisely defined spatially with 2p, leading to less extraneous background coming from outside of the illuminated region. Furthermore, the invisible near-infrared light used for 2p excitation avoids the visual responses triggered in the zebrafish by the visible 1p excitation[25], eliminating the "always-on" visually activated neurons that would otherwise yield background fluorescence that suppresses the detection of neurons undergoing spontaneous activity. This last point suggests that 2p-SVIM is best suited for studies of visually sensitive behaviors, such as brain-wide responses to visual stimuli (Supplementary Fig. 9) or sleep[26].

## Discussion

The results presented here demonstrate that, by combining the strengths of SPIM and LFM, SVIM provides a powerful tool for high-contrast, synchronous volumetric imaging of dynamic systems. By optimizing the illumination pathway, SVIM offers single-cell resolution, with improved contrast over wide-field LFM. The SVI principle was implemented in a recent work[27], where the sample was illuminated with a beam having a large cross-sectional area that filled up the volume of interest. This volume-filling strategy, compared with our volume-scanning strategy, is simpler to implement but gives up spatial precision in defining the volume of interest (Supplementary Note 1). SVIM is compatible and synergistic with recent innovations in LFM that optimize the detection pathway for more spatially uniform resolution and reduced grid-like artifacts in the reconstructions. These include implementation of multi-view light-field detection[27], and methods that capture and process light-field information through phase masks[28], diffusers[29], or in the Fourier domain[19,30]. SVIM could also be further optimized by approaches that speed up the image reconstruction or information extraction pipeline[20–22]. Together, these latest refinements of LFM and the high contrast of SVIM may enable LFM-based techniques to become the next-generation tools for imaging tissue-scale 3D dynamic biological systems. LFM-based methods belong to an emerging class of diverse computational imaging techniques[31,32] that harness the power of physical modeling, signal processing, and computation to enable new performance spaces beyond conventional microscopy. The selective volume illumination strategy of limiting the illuminated sample volume, in improving the contrast of the acquired image data, is positioned to play a key role in optimizing a variety of computational imaging approaches for a wide range of biological applications.

## Methods

**Microscopy setup and implementation.** The optical setup was based on an existing SPIM apparatus[7], with modifications to provide the selective volume illumination and light-field detection (Supplementary Fig. 1, Supplementary Table 2). Briefly, collimated beams from 1p excitation continuous wave and 2p excitation femtosecond-pulsed lasers were combined and directed at the sample through a pair of galvanometer scanners and scanning optics. The fluorescence signal was collected in the direction orthogonal to the illumination axis, through appropriate spectral optical filters, and directed to a detection module that allowed

imaging in either SPIM or SVIM mode. For SVIM mode, a micro-lens array was placed at the conventional image plane to capture the light field coming from the sample, which was subsequently recorded by the detection camera[3,4]. Computer-controlled motorized stages were used to allow reproducible switching between SPIM and SVIM modes. To provide selective volume illumination, the galvos controlling the illumination light were adjusted to paint out the desired illuminated volume multiple times within a single camera exposure. Image acquisition was through the software Micro-Manager[33] and custom software written in LabVIEW (National Instruments). See Supplementary Fig. 1 and its caption for more detailed descriptions, and Supplementary Table 1 for a list of key components of the microscopy setup. Supplementary Note 1 provides further discussion on the advantages and disadvantages of our volume-scanning implementation of SVIM.

**Sample handling and imaging procedure**. Zebrafish experiments: Fish were raised and maintained as described in ref. [34], in strict accordance with the recommendations in the Guide for the Care and Use of Laboratory Animals by University of Southern California, where the protocol was approved by the Institutional Animal Care and Use Committee (IACUC). All zebrafish lines used are available from ZIRC (zebrafish.org). Zebrafish embryos were collected from mating of appropriate adult fish (AB/TL strain) and raised in egg water (60 μg L$^{-3}$ of stock salts in distilled water) at 28.5 °C. At 20 hpf, 1-phenyl-2-thiourea (30 mg L$^{-1}$) was added to the egg water to reduce pigmentation in the animals. For imaging experiments, the samples were embedded in a 1-mm-diameter cylinder of 1.5% low-melting agarose (Sea Plaque) for imaging in the SPIM/SVIM setup, as described in ref. [7]. The imaging chamber was filled with 30% Danieau solution (1740 mM NaCl, 21 mM KCl, 12 mM MgSO$_4$•7H$_2$O, 18 mM Ca(NO$_3$)$_2$, and 150 mM HEPES) at 28°C. Anesthetic was used (buffered Tricaine, 100 mg L$^{-1}$) during the mounting procedure and imaging, except for during neural functional imaging. For structural imaging of the cranial vasculature, 5 days post fertilization (dpf) Tg(kdrl:eGFP) larvae were used. At this age, zebrafish larvae have not undergone sex differentiation. For heart imaging, transgenic adults, Tg(kdrl:eGFP) and Tg(gata1:dsRed), were crossed to produce offspring with fluorescent labels in both the blood cells (dsRed) and endocardium (eGFP). Samples, at 5 dpf, were selected for heterozygous expression in gata1:dsRed as this reduced the density of fluorescent blood cells. Two-color imaging of the blood cells and endocardium were carried out sequentially, at a rate of 90 volumes s$^{-1}$. For neural functional imaging, 5 dpf zebrafish larvae expressing the nuclear-localized pan-neuro fluorescent calcium indicators, Tg(elavl3:H2B-GCaMP6s), were imaged at a rate of 1 volume s$^{-1}$. Each bout of neural activity imaging lasted 150 s, with a visual stimulus, provided by a 625 nm light-emitting diode (LED; Thorlabs), turned on at $t = 100$ s.

Squid–bacteria experiments: Squid samples were procured as previously described[35]. Briefly, adult squids were collected in Oahu, Hawaii, and then housed and allowed to reproduce at a facility at the University of Wisconsin, Madison. Hatchling squid were shipped overnight to our laboratory, maintained in artificial seawater (Instant Ocean), and used for experiments within 3 days of arrival. A solution of 2% ethanol/artificial seawater was used to anesthetize the squid during mounting and imaging. Under a stereo dissection microscope, the mantle of the squid was carefully cut open and trimmed to expose the light organ of the animal. Then the animal was embedded in a cylinder of agarose (2% low melt agarose, 2% ethanol, in artificial seawater) using a procedure similar to that used for zebrafish. The cylinder and plunger unit were fashioned from a fluorinated ethylene propylene tube (inside diameter ~ 2 mm) and glass rod (outside diameter ~ 2 mm), respectively, to accommodate the size of the squid. To allow direct interaction between the light organ and the surrounding fluid environment, the small volume of solidified agarose encasing the light organ was carefully removed with forceps. The agarose-embedded squid was mounted into the imaging chamber (containing 60 mL of the ethanol/seawater solution), and GFP-expressing V. fischeri ES114 carrying pVSV102 (ref. [36]) was added to reach a concentration of 50,000 cfu L$^{-3}$. The squid and bacteria were monitored using SPIM for ~2 h before imaging of the bacterial flow fields with SVIM was carried out.

**Light-field image processing and 3D reconstruction**. Reconstruction of light-field images was carried out using the wave-optics procedure, described in refs. [4,5] and the software package made available by ref. [5]. Briefly, the rectifying parameters of the acquired light-field images, describing the geometrical relationships between the micro-lens array and the detection optical train, were found using the software LFDisplay[3]. Theoretical point spread functions (PSF) were calculated using the optical parameters of the entire imaging path, and the desired spatial sampling and coverage of the 3D reconstruction[5]. Then, the rectifying parameters and the PSF were used as inputs into the 3D wave-optics reconstruction program from ref. [5] to reconstruct the acquired 2D light-field images into 3D images. Two key parameters for the PSF calculation and the resulting 3D reconstruction are the z-extent of the volume to be reconstructed and the desired z-sampling (i.e., thickness of individual z-slice). Larger z-extent and finer z-sampling requires more onboard memory for the graphical processing unit (GPU) used in the 3D reconstruction program. With the GPU used here (Titan X, Nvidia), the largest z-extent that we could reconstruct was 400 μm, at z-sampling of 2 μm. Consequently, for results shown in Fig. 1d, e, and Supplementary Figs. 3 and 5, the datasets of both 550-μm SVIM and wide-field

LFM were reconstructed with z-extent of 400 μm; all others were reconstructed with z-extent equal to their actual experimental illumination extent.

We used the same micro-lens array (pitch = 150 μm, focal length = 3 mm) for two imaging conditions of (i) 32× magnification, 0.8 NA, and (ii) 20× magnification, 0.5 NA. With these parameters, and the reconstruction parameters listed in Supplementary Table 2, following[4] we expect the 32× magnification reconstructions to have ~3 (and 6) μm resolution laterally (and axially), and the 20× magnification reconstructions to have ~4 (and 12) μm laterally (and axially).

**Image analysis and presentation**. Raw images were background subtracted to account for camera dark counts, for both SPIM (done manually) and SVIM (done automatically in the software LFDisplay). All SPIM and SVIM images were scaled to fill the full 16-bit dynamic range. For visualization in the figures, unless otherwise noted, image pixel intensities were further scaled to minimum and maximum display contrast with 0.4% saturation. Unless otherwise noted, 3D images are presented as 2D averaged-intensity, instead of maximum-intensity, projections of 3D data in 2D format. Image processing and analysis in 2D were done in Fiji[37], while 3D rendering and analysis were done in Imaris (Bitplane).

Zebrafish vasculature: For zebrafish vasculature images (Fig. 1d, Supplementary Figs. 3–5), the 3D datasets were displayed as an averaged projection in z, for the same volume section extending from $z = -48$ to $-12$ μm, where $z = 0$ μm is the native focal plane of the detection objective. This volume excludes the native focal plane of the imaged z-stack, where grid-like artifacts from the light-field reconstruction are most prominent. The native focal plane was experimentally set at ~200 μm into the zebrafish head from its dorsal surface for the datasets shown in Fig. 1d and Supplementary Figs. 3–5.

Bacteria–squid: For squid-bacterial results (Fig. 2a–c, Supplementary Fig. 6, Supplementary Movies 1–3), tracking and quantification of the bacterial flow field were carried out using the automatic spot segmentation and tracking functions in Imaris, followed by manual correction.

Zebrafish heart blood: For results describing the zebrafish beating heart (Fig. 2d–g, Supplementary Fig. 7, Supplementary Movies 4–7), the sequentially acquired light-field time-series data of the endocardium and blood flow were reconstructed separately. The reconstructed four-dimensional (4D) datasets, each spanning approximately four heart beats, were then synchronized in time by renumbering the endocardium frames such that the time point at which the atrium is most contracted in the endocardium movie matches the time point when, in the blood flow movie, the flow into the ventricle from the atrium momentarily stops. After synchronization, the two movies were overlaid to create a composite two-color movie. Tracking and quantification of 12 representative blood cells' flow trajectories in the zebrafish heart were carried out manually in Imaris. We analyzed and presented the flow trajectories as they were directly derived from the manual tracking, to mainly demonstrate the benefits of SVIM. Follow-up work that aims to draw biological and biophysical insights from the imaged beating heart and blood flow should take into account the non-uniform resolution and image artifacts inherent with LFM in general.

Zebrafish brain activity: For the zebrafish brain activity results (Fig. 2h–m, Supplementary Figs. 8 and 9), we used an analysis pipeline based on spot segmentation of the time-domain standard deviation of the 3D time-series data to find active neurons[38]. First, from the light-field reconstructed 3D time-series data, we calculated the standard-deviation-projection along the temporal axis. The resulting time-projected 3D dataset was a spatial map of where the signal intensity changed substantially during the imaged time window, due to neuronal activity. Spot segmentation was then carried out on the time-projected 3D dataset using Imaris to find active neurons, with the constraint that neuronal nuclei appeared as ellipsoids with diameter of 5 and 10 μm in the lateral and axial direction, respectively, following the expected resolution of the light-field imaging and reconstruction. Once found, segmented neurons were used as spatial masks to extract the $\Delta F/F_0$ neural activity traces from the original reconstructed 3D time-series data. Spontaneous activity time window was from $t = 1$ to 100 s (Fig. 2h–m, Supplementary Fig. 8), while the visually evoked activity time window was from $t = 51$ to 150 s with the LED evoking light turned on at $t = 100$ s (Supplementary Fig. 9). For the latter case, k-means clustering of the activity traces was used to group the active neurons, identifying a subgroup that exhibited clear responses to the evoking light. Analysis was carried out using a combination of Fiji, Imaris, and MATLAB (MathWorks).

**Image contrast, simulated noise, and effective resolution**. Contrast: To compare the image contrast between SPIM, SVIM, and wide-field LFM, we measured the relative standard deviation of the pixel intensities from the respective images. The standard deviation σ of an image, which is the same quantity as root mean square contrast that appears in vision science[39,40], is given by

$$\sigma = \sqrt{\frac{1}{N-1} \sum_{i=1}^{N} (x_i - \bar{x})^2},  \qquad (1)$$

in which $x_i$ is a normalized intensity-level value, so that $0 \leq x_i \leq 65,535$ and $\bar{x}$ is the

average intensity of all pixel values $N$ in the image:

$$\bar{x} = \frac{1}{N} \sum_{i=1}^{N} x_i. \tag{2}$$

Putting the expressions above together, we have

$$Contrast = \frac{\sigma}{\bar{x}}. \tag{3}$$

This measure of image contrast is independent of the total pixel count $N$, and thus provides a concise metric for direct comparison of contrast between SPIM, SVIM with various SVI extents, and wide-field LFM. For the vasculature results (Fig. 1e, Supplementary Fig. 4f), we calculated the contrast for individual $z$-slices of the static 3D reconstructions to provide a $z$-depth-dependent comparison between different imaging modalities. To match the intrinsically lower axial resolution of SVIM and LFM (~12 μm), a moving average over 12 successive $z$-planes was applied to calculate the SPIM contrast curve. For the beating heart results (Supplementary Fig. 7c, f), the 4D reconstructions involved dynamic motion of the 3D samples, with substantial motion along the $z$-direction; therefore, we opted to calculate the contrast for the averaged-intensity $z$-projections at individual time points along the beating cycle. Analysis was performed in MATLAB.

Simulated noise: To simulate the extraneous background signal that would be expected from larger illumination extents, raw light-field images were corrupted with Poisson noise[41]. Simulated images were created by scaling up the raw 100-μm SVIM light-field image so that $\lambda$, the mean photon number of a pixel, would be equal to higher values. The raw 300-μm SVIM light-field image has $\lambda = 8400$, and larger illumination extents produce larger $\lambda$. Poisson noise was then applied to these scaled images on a per-pixel basis. The background noise produced is thus spatially correlated with the original raw 100-μm SVIM light-field image, which is a more realistic approximation than noise created from a uniform background. The resulting simulated images were then reconstructed and compared (Supplementary Fig. 4).

Effective resolution: To evaluate the effective resolution achieved with the various imaging modalities, we quantified the FWHM diameter of the same blood vessels captured by each modality (Supplementary Fig. 5, see its caption for full details). Briefly, starting in the SPIM 3D dataset, we selected a single $z$-slice that had four well-imaged blood vessels of approximately the same size. Matched $z$-slices in the SVIM and LFM datasets were found, and a MATLAB script generated line regions-of-interests across the selected blood vessels. Intensity line profiles were produced, normalized to a peak value of 1, from which the mean FWHM was calculated.

**Statistics and reproducibility**. No statistical tests were conducted.

**Reporting summary**. Further information on research design is available in the Nature Research Reporting Summary linked to this article.

## Data availability

Data underlying the plots in Figs. 2b and 4d–f are available as Excel files in Supplementary Data. All other relevant data are available from the authors upon request. Please send request to Thai Truong, tvtruong@usc.edu.

## Code availability

All relevant custom-written LabVIEW and MATLAB scripts/codes are available from the authors upon request. Please send request to Thai Truong, tvtruong@usc.edu.

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

## Acknowledgements

We thank E.G. Ruby (University of Hawaii-Manoa) for critical assistance and sharing of resources for the squid–bacteria experiments; Misha Ahrens (Janelia) for sharing of zebrafish lines; and Le Trinh for advice and support on zebrafish husbandry. Funding was provided for by the Gordon and Betty Moore Foundation grant #3396 (Ruby, McFall-Ngai and Fraser); National Institute of Health, grant #1R01MH107238-01 (Arnold, Kesselman, Fraser), grant #R37AI150661 (McFall-Ngai and Ruby), grant #R01OD11024 (Ruby and McFall-Ngai); National Science Foundation, grant #1650406 (Dickman, Fraser, Truong), and grant #1608744 (Kanso, Fraser). S.M. was supported by the USC Provost Fellowship; and K.K.D. by the Alfred E. Mann Doctoral Fellowship.

## Author contributions

T.V.T. conceived the idea, with further refinement from S.M., D.B.H., and S.E.F., T.V.T., D.B.H., S.M., A.A., J.V.T., M.J.M.N., and S.E.F. designed the experiments. T.V.T., S.M., D.B.H., and A.A. designed the microscopy setup. T.V.T. built the setup. T.V.T., D.B.H., S.M., A.A., and J.V.T. collected the data. S.M., D.B.H., D.E.S.K., A.A., and T.V.T. set up the light-field reconstruction pipeline. D.B.H., K.K.D., S.M., A.A., J.V.T., and T.V.T. analyzed the data and produced the figures. T.V.T., K.K.D., and S.E.F. wrote the manuscript, with inputs from all authors. T.V.T. and S.E.F. supervised the work. T.V.T., D.B.H., and S.M. contributed equally. A.A. and K.K.D. contributed equally.

## Competing interests

A patent application has been filed by the University of Southern California, with inventors T.V.T., S.M., D.B.H., snd S.E.F., for the methodology of SVIM. Application number: PCT/US2017/019512; pending. All other authors declare no competing interests.
