## [Peer Review File · Communications Biology]

Editorial Note: *This manuscript has been previously reviewed at another Nature Research journal. This document only contains reviewer comments and rebuttal letters for versions considered at Communications Biology.*

REVIEWERS' COMMENTS:

Reviewer #1 (Remarks to the Author):

I previously reviewed this manuscript and I am excited to see it again. I believe the authors clearly demonstrate the benefits of selective volume illumination, and nicely complement the work by Wagner et al and deserve publication. Specifically, they now compare quantitatively the benefits between SPIM, LFM, and SVIM. Additionally, the authors now more thoroughly characterize the performance of their system, including resolution, resolution uniformity, and also provide insight into the correspondence between voxel intensities in a SPIM and how they correlate with those measured by LFM. Furthermore, they show that the voxel intensities in SVIM are more accurate than LFM, which supports their selective illumination approach. All of these measurements will allow readers to evaluate the tradeoffs in LFM and SVIM and are likely to be very helpful in evaluating the merits of this approach. All of the minor things that I mentioned previously have been addressed, and I believe the paper is much more intuitive, concise, and informative. It is a powerful approach for imaging very fast timescale events with improved resolution and contrast. As such, I would recommend that the accept this manuscript for publication.

Reviewer #2 (Remarks to the Author):

The authors have made a thorough effort to address reviewers' comments, and I commend them on rewriting their article to fit very nicely into the brief communication format. It comes across very clearly and concisely. My remaining comments do not relate to the fundamentals of their technique and how it is presented, merely to the details of the demonstrations they have used. Nevertheless, I feel they should be addressed or clarified before final publication.

**** "Resolution" ****

I share reviewer #1's unease with the way resolution is defined and used in this manuscript. FWHM measures the size of an individual feature (and under some circumstances deconvolution-like processing can reduce a point source to a point source, irrespective of the diffraction limit of the imaging system). Resolution only has meaning when e.g. separating two nearby features, considering the Fourier spectrum, etc, and the question of resolution is particularly sensitive in imaging modalities where the "effective resolution" is highly sample-dependent.

I appreciate that resolution improvement is not the primary aim of this paper, but I am still uncomfortable with what I see as incorrect statements in relation to resolution, which is a concept with specific quantitative definitions. The biggest issue I have is with the statement "resulting in ~35% better resolution... (Supplementary Fig 5)" in the main text. SF5 shows a change in FWHM of a single isolated feature, and it is not ok to equate that to a quantitative statement of resolution improvement. To propose a constructive solution to my concerns, I would like to suggest to the

authors some additions qualifiers on their wording. I would like them to change that main text to ~35% reduction in FWHM (or equivalent wording as the authors wish). I would also like to see qualifiers such as "sample-dependent resolution", "nominal maximum resolution", "suggests improved resolution" etc. appearing where appropriate. I note that the authors seem to have chosen the wording of the final sentence in the SF5 caption quite carefully, and here they do not quite go so far as to claim a 35% improvement to resolution...

**** Neural activity ****

The authors compare results from WF LFM with 1p and 2p SVIM, but it is not entirely clear to me what how the differences should be interpreted.

- They state that SVIM "captur[ed] up to 4-fold more neurons" - is this a case of resolving additional nearby neurons that had been fused into one using WF LFM, or detecting more neurons that were previously lost in background noise, or something else?

- They refer to the "benefits of nonlinear excitation", but it is not clear to me what these benefits actually are, in the context of "painting" a volume. I would expect the familiar benefits of improved z contrast in 2p LFM to be largely irrelevant when painting a ~100um thick volume. Can the authors clarify?

- Related: the authors show differences (fig 2h-j) in the computed pixelwise temporal standard deviation. What is the origin of the apparent improvement from 1p to 2p excitation? Is this due to some genuine improvement in acquired image quality (and if so, how, as per my previous bullet?) or are the characteristics of the reconstructed images (effective resolution, etc) the same but the statistics of the neurons are genuinely different in 2p vs 1p?

**** Other queries ****

What is the origin of the apparent circular voids in fig 2g? Are they remnants of the "tracking blobs" that appear faintly in Suppl Movie 4?

I would expect ethical approval and specific licensing for research with 7dpf zebrafish, so presumably the authors should provide details in method 2, according to journal policy. All that is mentioned is a general statement about handling procedures.

Methods 4: "SVIM images were background-adjusted within the reconstruction process" - can the authors elaborate a little? Is this an automated procedure, or a manual process? How is the adjustment determined?

Suppl Fig 4: Can the authors clarify exactly how they introduced noise? I am not sure if they (A) simulated addition of a uniform background plus noise at a level of 8400 photons, or (B) simulated Poisson noise on their existing data on a per-pixel basis, i.e. added noise as a function of the original pixel value (with some particular mapping of pixel value to photon count). I presume they did (A), which seems to me to be the correct thing to do, but on first reading I understood it as (B). If I am correct, can the authors make the caption clearer?

Minor remark: I note to the authors that their response comment about block artefacts not affecting the centroid of an object is only true when the object is comparable to or larger than the block artefact size. To be fair that is probably true in their case here, though.

Responses to Reviewers

We thank both reviewers for their comments and suggestions.

We address the specific comments of reviewer #2 below. We include the reviewer's comments below, in blue, for reference.

The authors have made a thorough effort to address reviewers' comments, and I commend them on rewriting their article to fit very nicely into the brief communication format. It comes across very clearly and concisely. My remaining comments do not relate to the fundamentals of their technique and how it is presented, merely to the details of the demonstrations they have used. Nevertheless, I feel they should be addressed or clarified before final publication.

** "Resolution" **

I share reviewer #1's unease with the way resolution is defined and used in this manuscript. FWHM measures the size of an individual feature (and under some circumstances deconvolution-like processing can reduce a point source to a point source, irrespective of the diffraction limit of the imaging system). Resolution only has meaning when e.g. separating two nearby features, considering the Fourier spectrum, etc, and the question of resolution is particularly sensitive in imaging modalities where the "effective resolution" is highly sample-dependent.

I appreciate that resolution improvement is not the primary aim of this paper, but I am still uncomfortable with what I see as incorrect statements in relation to resolution, which is a concept with specific quantitative definitions. The biggest issue I have is with the statement "resulting in ~35% better resolution... (Supplementary Fig 5)" in the main text. SF5 shows a change in FWHM of a single isolated feature, and it is not ok to equate that to a quantitative statement of resolution improvement. To propose a constructive solution to my concerns, I would like to suggest to the authors some additions/qualifiers on their wording. I would like them to change that main text to ~35% reduction in FWHM (or equivalent wording as the authors wish). I would also like to see qualifiers such as "sample-dependent resolution", "nominal maximum resolution", "suggests improved resolution" etc. appearing where appropriate. I note that the authors seem to have chosen the wording of the final sentence in the SF5 caption quite carefully, and here they do not quite go so far as to claim a 35% improvement to resolution...

We agree with, and appreciate, the reviewer's overall suggestion of exercising caution and precision in discussing the concept of resolution. Following the reviewer's suggestion, we have modified the main text as follows:

- Main text, page 3, lines 6-7: "nominal maximum resolution", and "as approximated by the full-width-half-maximum (FWHM) of sub-diffractive fluorescent beads", are now used.
- Main text, page 3, lines 29-32: text is modified to now read "...resulting in better effective resolution even though SVIM and wide-field LFM utilize the same detection optics. This was

demonstrated in imaging $\sim 5 \mu\text{m}$ diameter blood vessels where SVIM achieved up to 35% improved FWHM over wide-field LFM....”

** Neural activity **

The authors compare results from WF LFM with 1p and 2p SVIM, but it is not entirely clear to me what how the differences should be interpreted.

- They state that SVIM “captur[ed] up to 4-fold more neurons” - is this a case of resolving additional nearby neurons that had been fused into one using WF LFM, or detecting more neurons that were previously lost in background noise, or something else?

We believe the detection of more neurons comes mainly from the reduced background of SVIM compared with wide-field LFM. This reduced background is demonstrated in Supplementary Fig. 8. We have modified the main text to highlight these points.

- Main text, page 4, line 34 to page 5, line 2.

- They refer to the “benefits of nonlinear excitation”, but it is not clear to me what these benefits actually are, in the context of “painting” a volume. I would expect the familiar benefits of improved z contrast in 2p LSFM to be largely irrelevant when painting a $\sim 100\mu\text{m}$ thick volume. Can the authors clarify?

- Related: the authors show differences (fig 2h-j) in the computed pixelwise temporal standard deviation. What is the origin of the apparent improvement from 1p to 2p excitation? Is this due to some genuine improvement in acquired image quality (and if so, how, as per my previous bullet?) or are the characteristics of the reconstructed images (effective resolution, etc) the same but the statistics of the neurons are genuinely different in 2p vs 1p?

We have modified the main text to clarify our descriptions of how the benefits of 2p excitation enable the improvement of 2p-SVIM over 1p-SVIM in capturing active neurons undergoing spontaneous activity. The relevant revised texts are duplicated below:

- Main text, page 5, lines 7-17: "...we found that 2p-excitation led to better contrast and a larger number of resolved active neurons (Fig. 4e,f, Supplementary Fig. 8). This detection of more active neurons, enabled by the improved contrast, results not only from the reduced tissue autofluorescence of 2p-excitation, but also from additional mechanisms specific to our application. First, 2p laser light preserves its spatial profile better than 1p light as it penetrates deeper into the sample⁹, due to the reduced scattering at longer wavelengths. Thus, the scanned selectively-illuminated volume is more precisely-defined spatially with 2p, leading to less extraneous background coming from outside of the illuminated region. Furthermore, the invisible near-infrared light used for 2p excitation avoids the visual responses triggered in the zebrafish by the visible 1p excitation²⁵, eliminating the "always-on" visually-activated neurons which would otherwise yield background fluorescence that suppresses the detection of neurons undergoing spontaneous activity."

**** Other queries ****

What is the origin of the apparent circular voids in fig 2g? Are they remnants of the “tracking blobs” that appear faintly in Suppl Movie 4?

They are indeed the "tracking blobs" from the image analysis, marking the cells whose trajectories were tracked and quantified. We have added the following sentence to the caption to describe this.

- Main text, page 12, lines 17-18, “Circular voids within several blood cells marked the cells whose trajectories were tracked and quantified.”

I would expect ethical approval and specific licensing for research with 7dpf zebrafish, so presumably the authors should provide details in method 2, according to journal policy. All that is mentioned is a general statement about handling procedures.

We made a mistake in our reporting that the zebrafish used in the neuroimaging experiment was 7dpf. They were 5 dpf. We have corrected this detail in the revised text.

Methods 4: “SVIM images were background-adjusted within the reconstruction process” - can the authors elaborate a little? Is this an automated procedure, or a manual process? How is the adjustment determined?

The background-adjustment procedure was automated, as part of the light-field 3D reconstruction pipeline provided by reference 5. The software set the lowest pixel value of each raw light field image to zero and the highest pixel value to 65535. We have revised the description in the manuscript to clarify this, copied below.

- Main text, Methods, page 8, lines 27-28, ““Raw images were background-subtracted to account for camera dark-counts, for both SPIM (done manually) and SVIM (done automatically in the software LFDisplay).”

Suppl Fig 4: Can the authors clarify exactly how they introduced noise? I am not sure if they (A) simulated addition of a uniform background plus noise at a level of 8400 photons, or (B) simulated Poisson noise on their existing data on a per-pixel basis, i.e. added noise as a function of the original pixel value (with some particular mapping of pixel value to photon count). I presume they did (A), which seems to me to be the correct thing to do, but on first reading I understood it as (B). If I am correct, can the authors make the caption clearer?

We introduced noise through a procedure closer to protocol (B) as described by the reviewer. We have revised the description in the Methods section to clarify the procedure and our intention:

- Methods, page 10, lines 25-32: "*Simulated noise*: To simulate the extraneous background signal that would be expected from larger illumination extents, raw light-field images were corrupted with Poisson noise⁴². Simulated images were created by scaling up the raw 100- μm SVIM light-field image so that λ , the mean photon number of a pixel, would be equal to higher values. The raw 300- μm SVIM light field image has $\lambda = 8,400$, and larger illumination extents produce larger λ . Poisson noise was then applied to these scaled images on a per-pixel basis. The background noise produced is thus spatially correlated with the original raw 100- μm SVIM light field image, which is a more realistic approximation than noise created from a uniform background. The resulting simulated images were then reconstructed and compared (Supplementary Fig. 4)."

Minor remark: I note to the authors that their response comment about block artefacts not affecting the centroid of an object is only true when the object is comparable to or larger than the block artefact size. To be fair that is probably true in their case here, though.